# Predictive Management Algorithm for Controlling PV-Battery Off-Grid Energy System

**DOI:** 10.3390/s21196427

**Published:** 2021-09-26

**Authors:** Tareq Alnejaili, Sami Labdai, Larbi Chrifi-Alaoui

**Affiliations:** Innovative Technologies Laboratory (LTI UR 3899), University of Picardie Jules Verne, 13 av. F. Mitterrand, 02880 Cuffies, France; tareq.alnejaili@albizia-technologies.com (T.A.); sami.labdai@u-picardie.fr (S.L.)

**Keywords:** energy management, forecasting, renewable energy, PV system, load side management, hybrid energy system

## Abstract

This paper introduces an energy management strategy for an off-grid hybrid energy system. The hybrid system consists of a photovoltaic (PV) module, a LiFePO4 battery pack coupled with a Battery Management System (BMS), a hybrid solar inverter, and a load management control unit. A Long Short-Term Memory network (LSTM)-based forecasting strategy is implemented to predict the available PV and battery power. The learning data are extracted from an African country with a tropical climate, which is very suitable for PV power applications. Using LSTM as a prediction method significantly increases the efficiency of the forecasting. The main objective of the proposed strategy is to control the different loads according to the forecasted energy availability of the system and the forecasted battery state of charge (SOC). The proposed management algorithm and the system are tested using Matlab/Simulink software. A comparative study demonstrates that the reduction in the energy deficit of the system is approximately 53% compared to the system without load management. In addition to this, the reliability of the system is improved as the loss of power supply probability (LPSP) decreases from 5% to 3%.

## 1. Introduction

The increase in using renewable energies can bring many advantages and opportunities; it can substantially contribute to the development of local communities. The expansion of renewable energy increases the reliability of energy systems and reduces the different impacts on the environment. In addition to this, the integration of renewable energy technologies can enhance energy security by adding variety to the overall electricity generation resources and reducing the risk of fuel spills [1].

Successful stand-alone systems typically take advantage of a combination of techniques and technologies to produce reliable energy, reduce costs, and minimize inconvenience. Some of these strategies involve the use of hybrid systems based on fossil or renewable fuels and/or reducing the amount of electricity required to meet energy needs.

Battery storage plays a basic role in stand-alone and stationary applications, and the improvement of the battery manufacture process and management can significantly increase the reliability of stand-alone energy systems [2]. 

The integration of renewable energy technologies requires adaptative control and management strategies due to the variable nature of the input energy (such as solar radiation and wind speed) [3]. 

Many research papers have studied various hybrid renewable power production structures. Jihane Kartite et al. [4] present a review of the different structures of hybrid systems in renewable energies. The main aim of their work was to present a brief explanation of the different renewable energy structures. 

Murugaperumal et al. [5] proposed an optimum design of a hybrid renewable energy system through load forecasting. Their results showed that the optimum energy configuration consists of three main renewable energy sources (PV, wind, bio).

Khanand Iqbal [6] introduced an investigation on a wind–hydrogen hybrid stand-alone power system. The main objective of their research was to test the system performance under variable stress such as sudden load variation and wind speed variation. 

Load energy management plays a very important role in the reliability of renewable energy systems; many research papers [7,8,9,10,11] deal with different energy management strategies. Alnejaili et al. [12] present the dynamic control and advanced load management of a stand-alone hybrid renewable power system for remote housing. The main aim of their work was to control the power flow and optimize the operation of the energy system.

Alnejaili et al. [13] present a new energy management strategy for a (PV–wind–diesel) hybrid system. Their results show that using the proposed strategy can reduce the peak consumption of the system.

Liu et al. [14] introduced a load-adaptive real-time energy management strategy for a battery/ultracapacitor hybrid energy storage system. Their results demonstrated the effectiveness of the real-time management on the energy balance of the power system.

The authors of [15] proposed the optimal sizing of a PV and battery energy storage system (BESS) for a grid-connected house. Their management algorithm was based on the flat and time-of-use (TOU) electricity rate in Australia. Their comparative study showed that the hybrid battery–PV system yields better performance in comparison with PV systems. In [16], the authors proposed a peak shaving algorithm for a micro-grid equipped with a BESS. Their proposed algorithm could shift the peak demand of the day based on a decision tree algorithm. Their results illustrate the cost benefit of their proposed peak shaving method in comparison with conventional peak shaving methods.

Control and management systems that depend on power forecasting are gaining greater significance nowadays. They allow us to achieve a better decision-making and management strategy. The forecasting is greatly important with renewable energy-based systems, which are strongly affected by the metrological conditions and the availability of these different energy sources.

Dongho Lee et al. [17] used linear regression and classical methods for short-term load forecasting for energy management systems; their study was based on small- and medium-sized office buildings in South Korea.

The authors of [18] presented a review on the topic of Intelligent Systems for Power Load Forecasting; different algorithms and methods were implemented in the literature on load profile forecasting. They came to the conclusion that neural networks and fuzzy set predictions give better results in the field of power and load forecasting than the other methods.

The authors of [19] have compared ARIMA-based methods with recurrent network-based algorithms; they also concluded that neural networks yield better performance than classical methods.

The authors of [20] presented an electricity consumption framework based on an Adaptive Neural Network Inference System (ANFIS). Their ANFIS algorithm was configured using a Multi Objective Genetic Algorithm, and their results were validated in a real user-side context with real load changes. V.A. Boicea in [21] presented a medium-term load forecast (MTLF) at power system level, based on the Big Data concept and Convolutional Neural Networks (CNNs). However, ref. [12,13,14,15,16] did not include a prediction of the produced power or a prediction of the batteries’ SOC, which are important factors in our proposed algorithm.

The main aim of this work was to enforce the capability of battery protection and energy-saving under unknown load patterns. We introduce a stand-alone hybrid power system that consists of a PV panel as the main energy source and a LiFePO_4_ battery pack for energy storage; the management algorithm will take into consideration the meteorological conditions (temperature and lightning), the occupancy of the building, and the forecasted production and battery state of charge during the next day.

A Long Short-Term Memory network (LSTM) is used as a forecasting method. Using LSTM as a prediction method can significantly increase the efficiency of the forecasting and enhance the overall system efficiency. The system and the management algorithm are tested by the means of simulation using Matlab/Simulink, and the simulation results demonstrate the efficiency and security improvement of the system.

The rest of this paper is divided into five main parts. In Section 2, we illustrate the system; then, the system modeling is presented. We present the management algorithm, and in Section 3, we present the forecasting and power management results. The discussion is presented in Section 4; finally, the conclusions are presented in Section 5.

## 2. Materials and Methods

### 2.1. System Description

The proposed configuration of the energy system shown in Figure 1 is composed of a PV module and a LiFePO4 battery pack coupled with a Battery Management System (BMS). The BMS ensures the safe operation of the battery; it protects the battery against the overcurrent, the over-temperature, and the overvoltage. In addition to this, it monitors the battery cell’s voltage and communicates with the battery charger.

The PV is connected to a hybrid solar MPPT inverter that controls the battery charging and discharging process.

A central load management unit is added to the system, which controls the AC outputs of the system and communicates with the BMS and the solar inverter.

The different load types are connected to the AC bus via relays to manage the loads depending on the energy state of the system. The studied system characteristics are given in Table 1.

### 2.2. Modeling

#### 2.2.1. Modeling of the Photovoltaic Generator

The PV power is modeled according to the following equation [13,22]:(1)$$P_{PV}=\eta_{p}.G.S_{PV}$$
where $\eta_{p}$ is the instantaneous efficiency of the *PV*, *G* is the irradiance (w/$m^{2})$, and $S_{PV}$ is the surface area of the *PV* module. The $\eta_{p}$ can be expressed as:(2)$$\eta_{p=}\eta_{r}.\eta_{pt}\left[1-\beta_{p}\left(T_{M}-25\right)\right]$$
where $\eta_{r}$ is the efficiency of the module at the reference temperature, $\eta_{pt}$ is the power tracking efficiency, *βp* is the thermal efficiency coefficient of the PV generator material, and $T_{M}$ is the temperature of the cell and can be expressed as:(3)$$T_{M}=T_{A}+\frac{NOCT-20}{800}.G$$
where $T_{A}$ is the ambient air temperature and NOCT is the nominal operating cell temperature. Table 2 summarizes the parameters used in the PV modeling.

#### 2.2.2. Inverter Modeling

The inverter can be modeled according to its output power as follows [9]:(4)$$P_{ac}=\eta .P_{dc}$$
where $P_{dc}$ is the inverter input power and *η* is the inverter efficiency and can be expressed as [9,12].
(5)$$\eta =\frac{\left(\frac{P_{ac}}{P_{imax}}\right)}{k_{0}+\left(1+k_{1}\right)\left(\frac{P_{ac}}{P_{imax}}\right)+k_{2}{\left(\frac{P_{ac}}{P_{imax}}\right)}^{2}}$$
where $k_{0}$ is the no load loss coefficient, $k_{1}$, $k_{2}$ are linear and quadratic current loss coefficients, and $P_{imax}$ is the inverter maximum input power. Table 3 summarizes the parameters that are used in the mathematical modeling of the inverter.

#### 2.2.3. Battery Energy Modeling

The instant power of the battery can be expressed as [10,22]:(6)$$P_{battery}\left(t\right)=P_{PV}\left(t\right)-Pload\left(t\right)$$
where Pload(t) is the instant load power. The battery state of charge (SOC) can be expressed as [23,24]:(7)$$SOC\left(t\right)=SOC\left(t-1\right)+P_{battery}\left(t\right)/P_{battery\_nominal\;}$$
where $P_{battery\_nominal\;}$ is the nominal power capacity (Wh) of the battery.

### 2.3. Power Flow Control and Management

#### 2.3.1. Power Control

The central controller is designed to manage the energy flow of the system. Indeed, it controls the battery charging–discharging cycles and introduces a second security level for the LiFePO4 battery. Figure 2 presents the operation strategy used in the central controller and can be described as follows.

If the power generated by the PV is less than the load power, the difference will be provided by the battery. If the battery state of charge (SOC) is below its minimum level, the different loads will be disconnected.

In case of power excess, the surplus power will be transferred to the battery. In fact, if the battery SOC is higher than its maximum level, the power excess will be used to power a water pump. In this way, the surplus or lost energy will be highly limited.

#### 2.3.2. Load Management Strategy

The main objective of the load management strategy is to optimize the operation of the solar energy system and reduce any power deficit. The AC load is divided into three main types depending on their priorities:Loads type 01: These are the loads that have priority over the other load types. They include the different lamps and the refrigerator.Loads type 02: These loads are less important than the load type 01 and include fans and TV.Loads type 03: This load is mainly a water pump that will be used to redirect any energy excess, which increases the energy efficiency of the system.

The proposed management strategy takes into consideration the following inputs:The ambient temperature (Ta);The natural irradiance (Ga);The occupancy—$O_{C}=1$ if the house is occupied and $O_{C}=0$ if not;The predicted battery state of charge (*PSOC*) at time *t* + Δ*ta*;The predicted PV generated energy PEpv at time *t* + Δ*ta*.

The three inputs Ta, Ga, and $O_{C}$ are measured directly in real time via their appropriate measurement tool, and the predicted data are based on their previous values only and are independent of time. The load management algorithm is presented in Figure 3.

The refrigerator is the most important load and its operation is subject to the following condition:PSOC(t+Δts) ≥ $SOC_{min}$

The control of the different lamps is subjected to the manual control; they are controlled according the occupancy and the irradiance and can be expressed as follows:$(O_{C}$ = 1) & ($G_{a}$ ≤ $G_{aref}$)

The control of the Tv is subject to the following conditions:$\frac{PE_{PV\;}\;\left(t+\Delta ta\right)}{E_{PV\;\left(nominal\right)}}*100\%\;>PE_{PV\_security}\;\&\;PSOC\left(tr\right)>SOC_{security}$

The first condition is that the PV generated energy during the next day must be greater than a pre-selected limit ($PE_{PV\_security})$. This parameter and the nominal energy production $E_{PV\;\left(nominal\right)}$ can give the control system a good view of the availability of the solar energy during this time period.

The second condition is that the predicted battery SOC at tr = 18 h must be greater than a pre-selected battery SOC limit. This parameter at this specified hour can give the control system a perfect illustration of the energy availability of the system, as the load peak consumption occurs at night.

Along with the other limits (Table 4), the security limits ($PE_{PV\_security},SOC_{security})$ are chosen to guarantee the continuous operation of the critical loads and to avoid any power deficit. In addition to this, applying these limits will increase the service life of the battery as the DOD of the battery will be limited.

The fan operation is controlled according to the ambient temperature and occupancy as follows:$T_{a}$ ≥ $T_{aref}$ & *Oc* = 1

The water pump will be enabled according to the prediction of the power excess; this condition can protect the battery lifetime and prevents overheating:PSOC(t+Δts) ≥ $SOC_{max}$

#### 2.3.3. LSTM and Forecasting Algorithm

The LSTM networks are a type of recurrent neural network that has the ability to cope with hard nonlinear approximation tasks of time series. They are far superior to recurrent neural networks and classical forecasting methods that depend on time series analysis. Due to their specific internal architecture, they are very reliable in short-term and long-term forecasting without having to deal with the vanishing gradient problem [14].

The internal structure of an LSTM network is composed of many perceptrons that are connected in a well-organized manner, as shown in Figure 4. Each perceptron can be modeled as a gate, and these gates can feed forward the information contained in them; they are the input gate, output gate, and cell gate. Another type of gates is called the forget gates, which have the ability to remove information from the next time step output. The gates are controlled via sigmoidal() and tanh() activation functions. Another important element is the cell state, which is directly connected to the LSTM output, and the other gates are used to control the information flow to this cell.

The equations that describe the relations between the output of the LSTM, the cell, and the different gates can be expressed as follows [19,25]:(8)$$\{\begin{array}{c} a_{t}=\sigma \left(w_{a}*x_{t}+u_{a}*Y_{t-1}+b_{a}\right)\\ i_{t}=\tanh\left(w_{i}*x_{t}+u_{i}*Y_{t-1}+b_{i}\right)\\ f_{t}=\sigma \left(w_{f}*x_{t}+u_{f}*Y_{t-1}+b_{f}\right)\\ o_{t}=\sigma \left(w_{o}*x_{t}+u_{o}*Y_{t-1}+b_{o}\right)\\ {\mathrm{state}}_{t}=a_{t}*i_{t}+f_{t}*{\mathrm{state}}_{t-1}\\ Y_{t}=\tanh\left({\mathrm{state}}_{t}\right)*o_{t} \end{array}.$$
where $x_{t}$ is the LSTM inputs, $a_{t},i_{t}$ is the input gates, $F_{t}$ is the forget gates, $\;g_{t}$ is the output gates, $\;state_{t}$ is the cell state, and the LSTM output is $Y_{t}$, $\;\;state_{t}$ is the cell state, σ is the sigmoidal activation function, $w=\left[w_{a}\;w_{i}\;w_{f}\;w_{o}\right]$ are the network feed forward weights, $u=\left[u_{a}\;u_{i}\;u_{f}\;u_{o}\right]$ are the network recurrent weights, and $b=\left[b_{a}\;b_{i}\;b_{f}\;b_{o}\right]$ are the network bias weights.

## 3. Simulation Results

The main objective of the long-term simulation process is to explore the behavior of the energy system under various stresses. The effectiveness of the proposed energy management strategy will be evaluated in terms of loss of power supply probability (LPSP).

### 3.1. Application Site and Load Profile

Figure 5 shows a typical load profile of a selected location (an African tropical country with a high-quality natural irradiance). It is characterized by a peak consumption during the night period. It has 160 W of peak power consumption and 1.475 kWh/day of daily energy consumption.

The simulation is performed over one year and uses the collected weather data of the selected location. The weather data are obtained using a Photovoltaic Geographical Information System (PVGIS). Figure 6 shows the monthly production of the PV panels, while Figure 7 shows the monthly load consumption.

### 3.2. Forecasting Results

Figure 8 presents the forecasted PV power (Figure 8a) and the battery SOC (Figure 8b). We used the well-known LSTM neural network prediction; with our technique, we can predict 24 h from the data of four previous consecutive days.

A more general view of the SOC and PPV forecasted values over 25 days in the future is presented in Figure 9a and Figure 10a. In Figure 9b, we present the PSOC(tr) at tr = 18 h for each subsequent day. In Figure 10b, we present the value of the total generated power on the next day $PE_{PV\;}\;\left(t+\Delta ta\right)$. These values will be critical in our decision-making algorithm.

In Figure 11a,b, we present a yearly histogram of the LSTM forecasting error of the SOC and PPV, respectively.

### 3.3. Management Algorithm Results

With the results, we conduct a comparative study between the system with load management and without load management. Figure 12a shows the load cutoff duration and Figure 12b shows the energy deficit of the system. Figure 13 shows the monthly average state of charge of the battery.

## 4. Discussion

From Figure 8, Figure 9, Figure 10 and Figure 11, we can clearly see that the forecasting error is generally around zero, with inevitable small errors.

Figure 12a shows that the load cutoff duration is highly decreased using the proposed load management. The total cutoff duration of the system without load management is 366.7 min (15.25 days), while it is equal to 169.6 (7 days) minutes in the case of the system with energy management.

The system with load management reduces the energy deficit of the system. The reduction is approximately 53% compared to the system without load management; see Figure 12b.

Figure 13 shows that the monthly average state of charge of the system is considerably improved using the proposed management strategy. In addition to this, the reliability of the system is improved as the LPSP decreases from 5% to 3%.

## 5. Conclusions

This paper presents an energy management strategy for an off-grid (PV battery) energy system. Its main objective was to control the different loads according to the forecasting of the energy availability of the system and the prediction of the battery SOC at peak hour and the total power to be delivered the next day by the PV panels. Finally, the forecasting results are encouraging and the LSMT network shows its efficiency in tracking the future time series data. The proposed system has been tested using Matlab/Simulink software. The result demonstrates the efficiency of the control algorithm: it reduces the energy deficit of the system, and the decrease was around 53% compared to the system without load management. In addition to this, the reliability of the system is improved as the LPSP decreases from 5% to 3%.

## Figures and Tables

**Figure 1 sensors-21-06427-f001:**
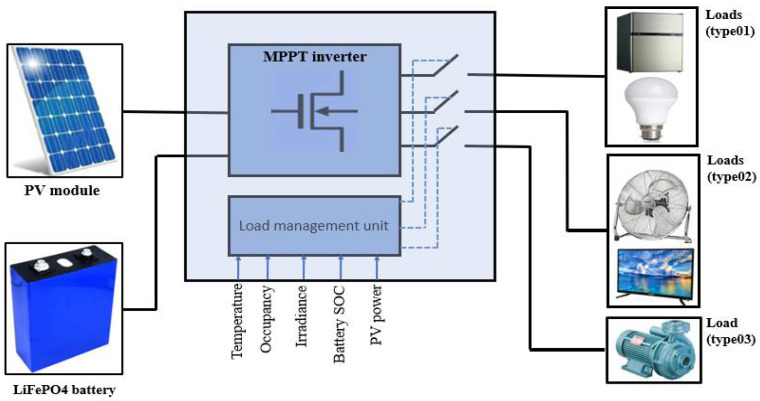
System configuration.

**Figure 2 sensors-21-06427-f002:**
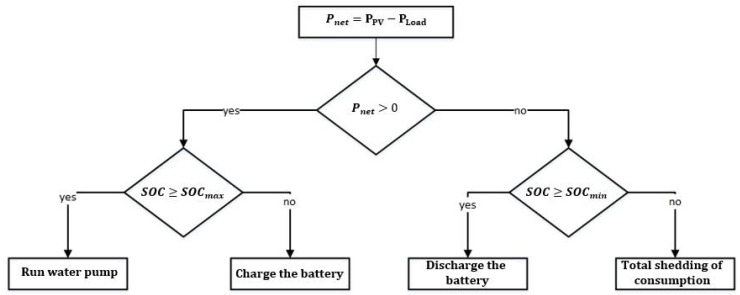
Operation strategy of the power system.

**Figure 3 sensors-21-06427-f003:**
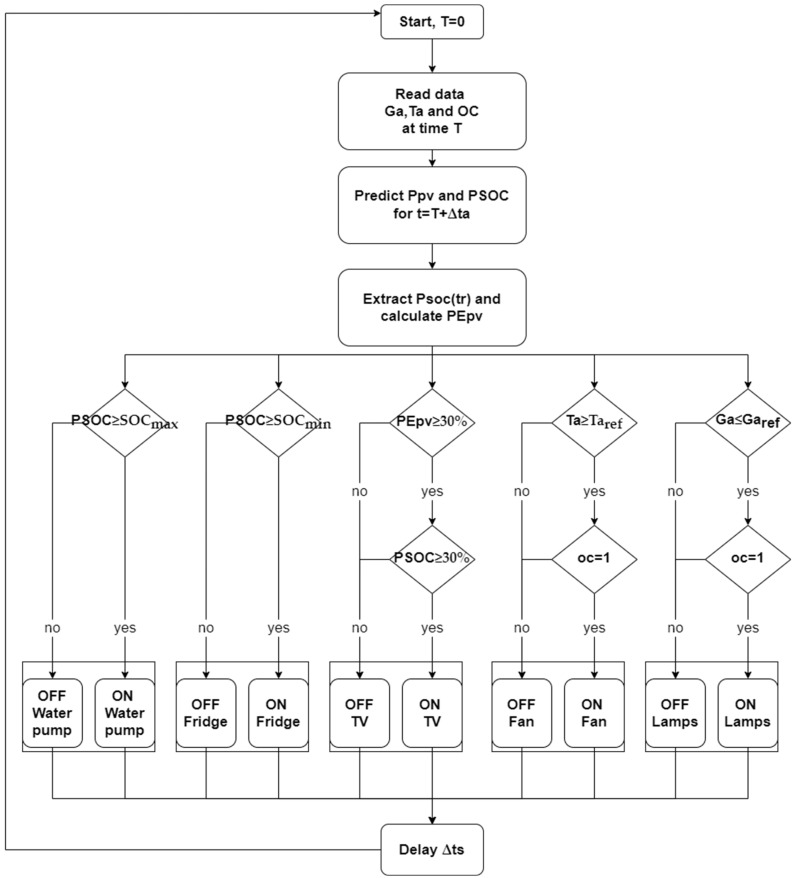
The load management strategy flowchart.

**Figure 4 sensors-21-06427-f004:**
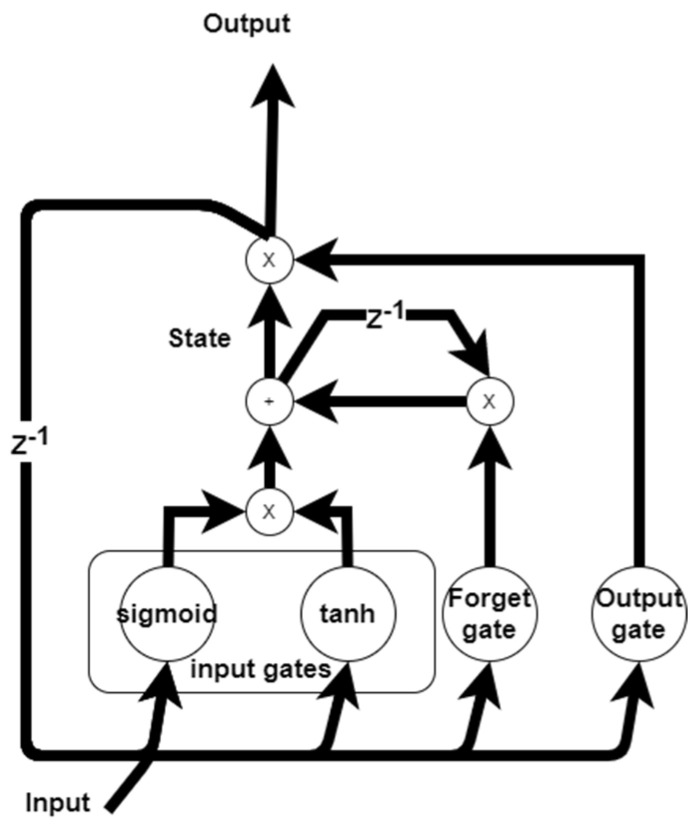
LSTM internal structure.

**Figure 5 sensors-21-06427-f005:**
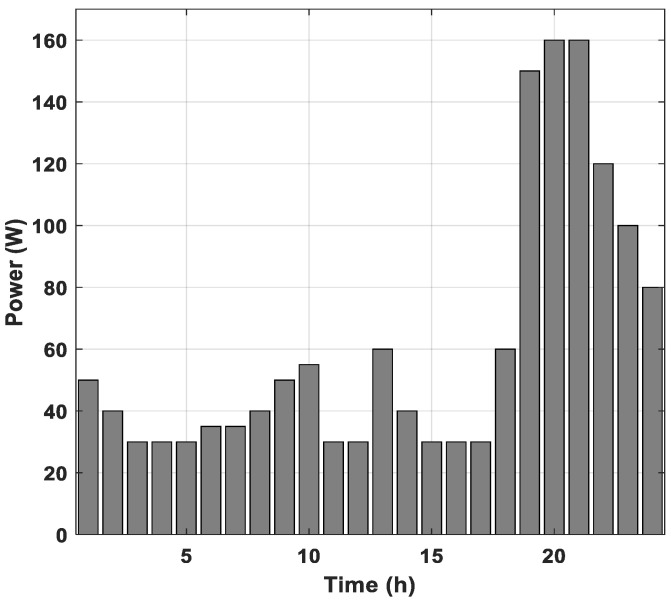
Daily load profile.

**Figure 6 sensors-21-06427-f006:**
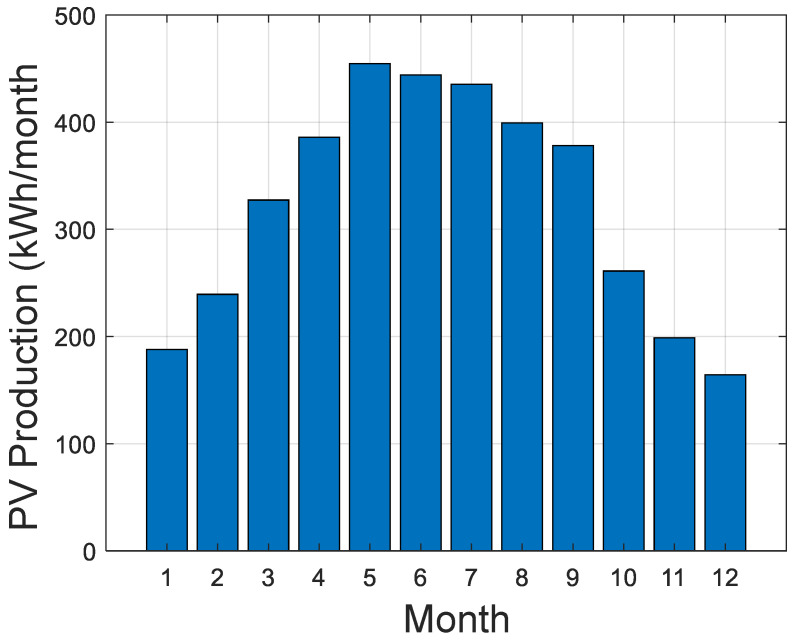
PV monthly production.

**Figure 7 sensors-21-06427-f007:**
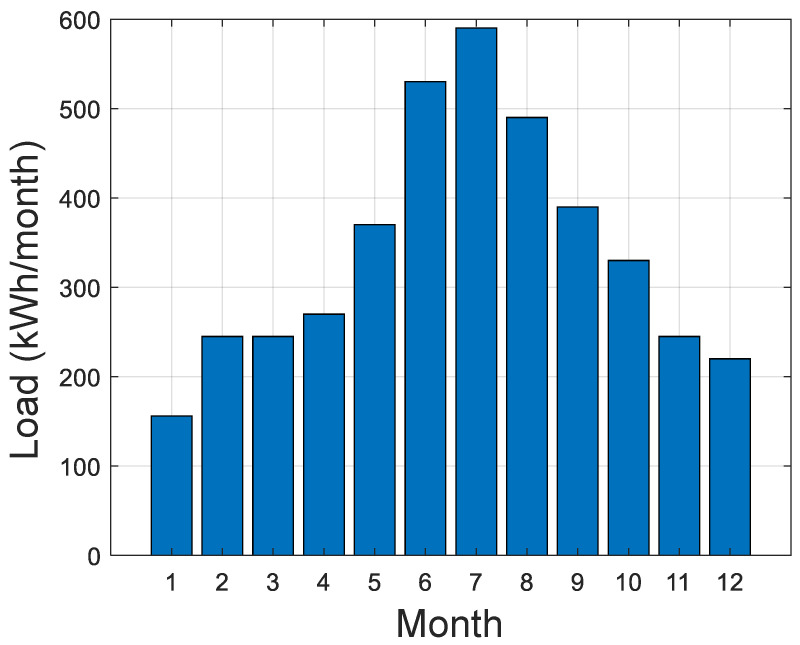
Load monthly consumption.

**Figure 8 sensors-21-06427-f008:**
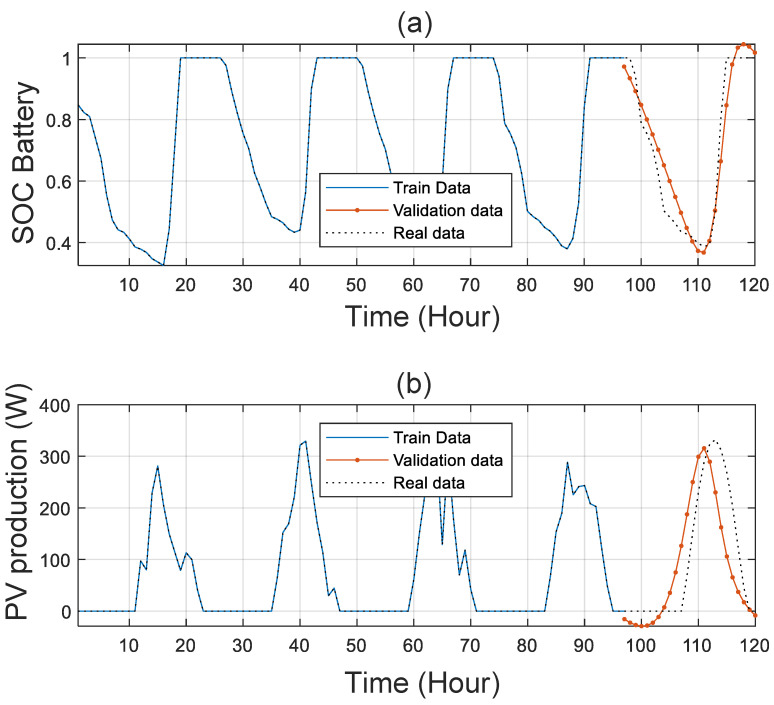
(**a**) Hourly state of charge. (**b**) Hourly production of the PV panels.

**Figure 9 sensors-21-06427-f009:**
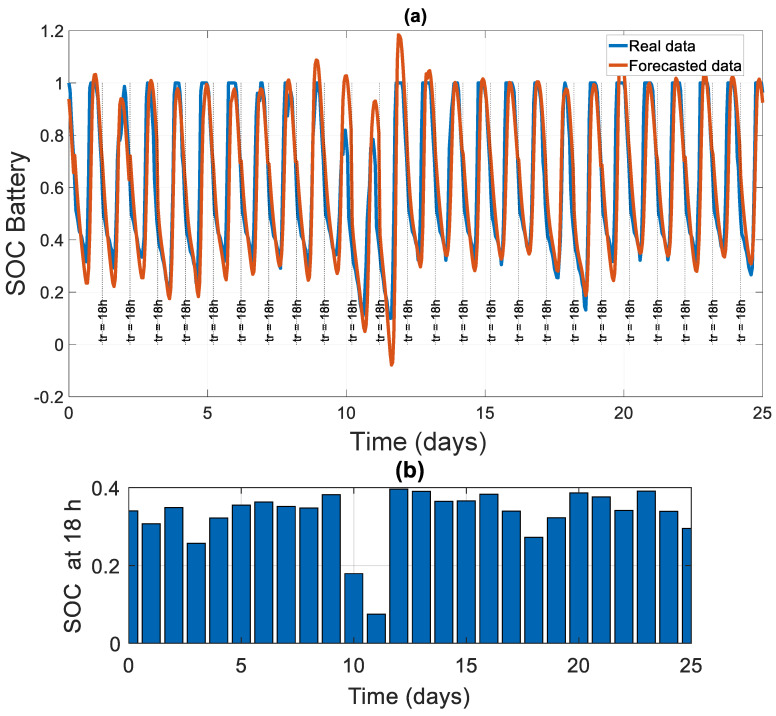
Battery SOC forecasting results over 25 days using LSTM network: (**a**) battery SOC real and forecasted data; (**b**) forecasted battery PSOC at tr = 18 h00.

**Figure 10 sensors-21-06427-f010:**
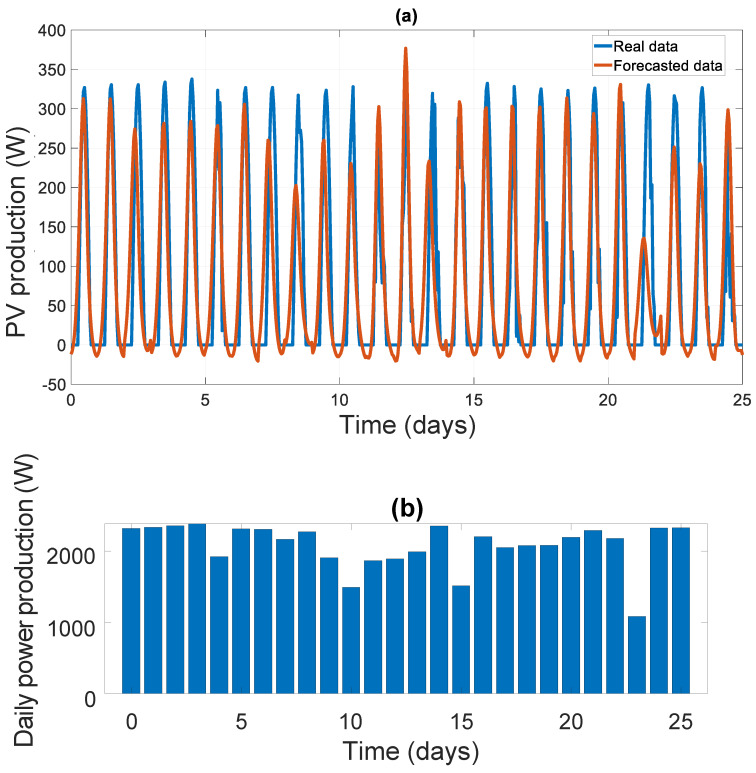
PV panel power forecasting results over 25 days using LSTM network: (**a**) PV panel power real and forecasted data; (**b**) forecasted PV panel power at the next day.

**Figure 11 sensors-21-06427-f011:**
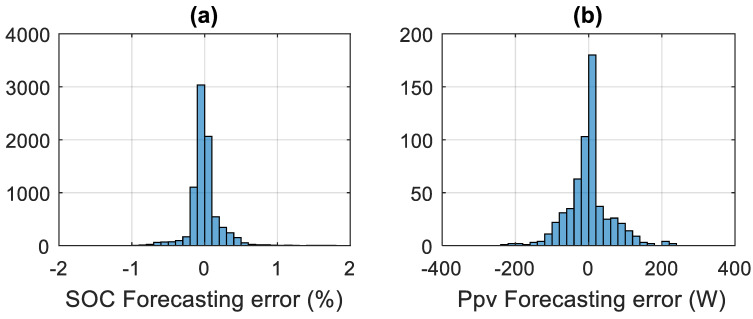
Yearly forecasting error: (**a**) histogram of PPV power forecasting error; (**b**) histogram of battery SOC forecasting error.

**Figure 12 sensors-21-06427-f012:**
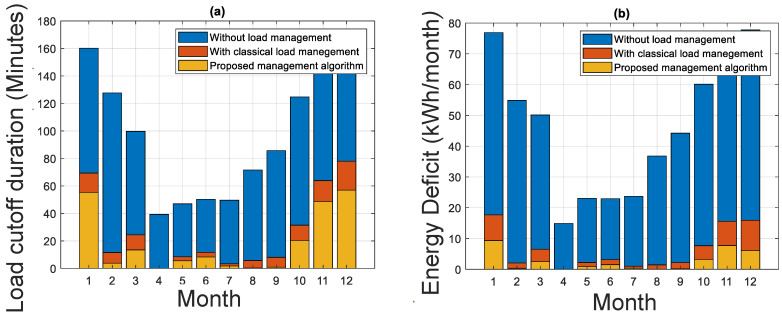
(**a**) Load cutoff duration, (**b**) energy deficit.

**Figure 13 sensors-21-06427-f013:**
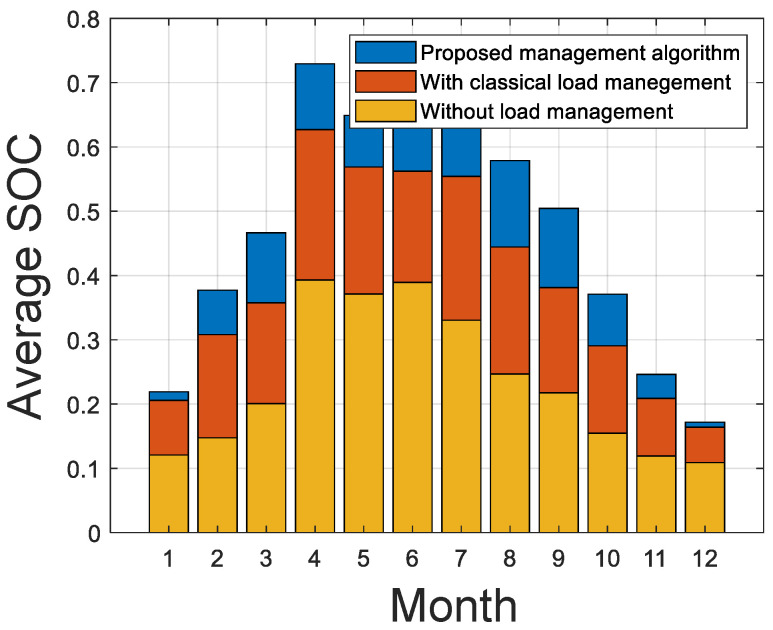
Battery SOC.

**Table 1 sensors-21-06427-t001:** System characteristics.

Component	Characteristics
Battery	1300 Whr
PV	400 Wc
Inverter	1 kW

**Table 2 sensors-21-06427-t002:** Parameters of the photovoltaic system.

Symbol	Value
$S_{PV}$	2.3 m^2^
$\eta_{r}$	16%
$\eta_{pt}$	0.98
$\beta_{p}$	0.5%/°C
NOCT	25 °C

**Table 3 sensors-21-06427-t003:** Inverter simulation parameters.

Symbol	Value
η	95%
$P_{imax}$	1 kW
$k_{0}$	0.005
$k_{1}$	0.005
$k_{2}$	0.06

**Table 4 sensors-21-06427-t004:** Parameters of the management system.

Constant	Value
$SOC_{min}$	10%
$PE_{PV\_security}$	30%
$SOC_{security}$	30%
$SOC_{max}$	95%
$T_{aref}$	20 °C
$G_{aref}$	100 W/m^2^

## Data Availability

Not applicable.

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
