# Peer review of "Predictive Management Algorithm for Controlling PV-Battery Off-Grid Energy System"

_sensors, 2021, doi:10.3390/s21196427_

Round 1

Reviewer 1 Report

The topic is interesting and it is adapt to this journal. The collaboration among several faculties is useful and I think that there is a great work behind the presentation of this work. The manuscript and the research in it are well structured. However, in my opinion, the paper is sometimes difficult to follow and more information is required on some issues. My comments:

-Clarify better the innovation of this work in the abstract and in the main text.

-Read articles to understand the structure of Sensors. The following structure would be preferable based on the Sensors Microsoft Word template file: 1. Introduction (1.1, 1.2, 1.3.), 2. Materials and Methods (2.1, 2.2., 2.3.), 3. Results (3.1, 3.2, 3.3), 4. Discussion (4.1, 4.2, 4.3), 5. Conclusions.

-Please search references to all equations. Equations should always be accurately and clearly referenced. These are missing in several places (e.g. Chapter 3.3.).

- The y-axes of Figure 8.b and Figure 10.a are missing the unit of measure.

-Please add more information's about the model validation. The validation process needs to be described more deeply.

-Extend the conclusion with more general usability. What are the benefits of the results in a global context? Please explain this better in the manuscript.

-At the end of the study need to create a nomenclature table with units.

Author Response

Dear reviewer,

Reviewer 2 Report

Dear Authors,

Thank you for taking the time to run the experiments, and preparing the manuscript. 

Please see my inputs below:

Language: The English language used in this manuscript meets the expected standard of scientific communication.

Comments/suggestions/questions:

Abstract:
It is brief as expected, and conveys the contribution of this work in an effective manner.

Introduction:
The literature survey covers the key areas/works of the field. The beginning paragraphs may be enhanced (see my suggestions at the end).

Section#2:
The image quality and presentability should be improved.

Section#3:
Modeling descrives the involved components in proper way.

Section#4:
Naming should be consistent. Line #72 uses 'lifepo4', and line#152 uses 'lifpo4'
Please use more commonly seen format: LiFePO4
Figure#2 should be redrawn with proper subscripts. In this age, it is expected that the figures are also drawn carefully and with proper care (without using SOC_min to denote subscript in image).
Authors should present reasoning for line #190. The reason for selecting 30% needs to be mentioned.

Figure#3: Better quality graphics/text expected.

Line #235: 'atypical' should be 'a typical'. Also, "(African tropical country with a high-quality natural irradiance)", should be "(an African tropical country with a high-quality natural irradiance)"

Section#5:

Line#244: Please refer the source of weather data.
Figures in the remaining sections are well presented. Please place figure 12.a and 12.b side by side (two columns).

Line#282: It is referring to figure #14 which does not exist. Please fix the typo to refer to figure #13.

Suggestion: The beginning part of the Introduction may be improved with more discussion of overall picture and the domain which this manuscript will be touching on, with proper references. For example, energy security (line#25):https://www.mdpi.com/1996-1073/14/15/4639 ;cost reduction of batteries and overall system(line#28) https://www.mdpi.com/2313-0105/3/2/17; hybrid energy(line#29): https://www.mdpi.com/1996-1073/14/13/3928, etc.

Once again, thank you for your work in this interesting topic, and I recommend this for publication with the above revisions.

Sincerely
The Reviewer

Author Response

Dear reviewer,

Reviewer 3 Report

The paper proposed a predictive algorithm to control an off-grid system. The paper focuses an interesting area of research. However, paper need significant improvement in the following areas:

  • Novelty/Contributions of the paper is not clearly defined. Authors need to undertake a rigorous literature review in order to explore the research gaps in the research domain. From which authors can identify the research contributions of the paper. Few of the research authors may explore:
  • Optimal Sizing of Rooftop PV and Battery Storage for Grid-Connected Houses Considering Flat and Time-of-Use Electricity Rates
  • A novel peak shaving algorithm for islanded microgrid using battery energy storage system
  • There are many research that deals with energy management strategy in off-grid microgird systems that minimises energy costs. It is great that authors have used prediction algorithm. However, authors have not explore existing researches that proposed prediction model in the similar research area.
  • A comparative analysis with the current researches is essential to identify the effectiveness of the proposed approach.
  • A step by step methodological flow chart will be great.
  • More attention is required in finalysing the flow chart as few of the links/connection/flow looks confusing.
  • Authors may compare their results with other available algorithms.

Author Response

Dear reviewer,

Round 2

Reviewer 3 Report

The authors updated the paper addressing reviewers comments. However, the paper may improve further to address reviewers comments critically. Moreover, figure quality need to be improve for better visibility of the paper.
